# Susceptibility to Postoperative Changes in Music Appreciation in Elderly Cochlear Implant Recipients

**DOI:** 10.3390/jcm11175029

**Published:** 2022-08-27

**Authors:** Jee-Hye Chung, Min-Kyu Kim, Da Beom Heo, Jong Bin Lee, Jin Woong Choi

**Affiliations:** 1Department of Otorhinolaryngology-Head and Neck Surgery, Chungnam National University College of Medicine, 282 Munhwa-ro, Jung-gu, Daejeon 35015, Korea; 2Department of Otorhinolaryngology, Konyang University College of Medicine, Daejeon 35365, Korea

**Keywords:** cochlear implant, music appreciation, elderly, age effect, questionnaire

## Abstract

With the rise in life expectancy and the consequent increase in the elderly population, the use of cochlear implants (CI) in elderly patients with hearing loss is also increasing. The aim of this study was to investigate whether music appreciation in elderly CI users differs from that of non-elderly users. Forty-nine adult CI recipients participated in the study, and the Korean version of the Music Background Questionnaire was utilized preoperatively and postoperatively to evaluate music appreciation. The changes between the preoperative and postoperative values were compared after categorizing the participants into a non-elderly group (<65 years; *n* = 31) and an elderly group (≥65 years; *n* = 18). When compared to the non-elderly group, the elderly individuals exhibited a significant decrease in music listening times, without a significant change in the genre of music listened to following CI surgery. Moreover, the elderly group demonstrated significant decreases in music appreciation scores in terms of music quality and music elements, perceiving music as less natural, less clear, and more complex. They also exhibited significant changes in scores with respect to perception of rhythm, melody, timbre, and lyrics. This susceptibility to postoperative changes in music appreciation among elderly CI users should be considered in surgical counseling and music training programs.

## 1. Introduction

The use of a cochlear implants (CIs) is an effective hearing rehabilitation method for bilateral severe-to-profound hearing loss for cases in which a hearing aid is no longer effective. The improved hearing and speech perception facilitated by a CI not only allows prelingually deaf children to develop their language capacity, but it also helps postlingual hearing loss patients improve their quality of life [1,2,3]. However, the efficacy of implants is limited in some complex listening environments; for example, in situations involving conversations with several people with or without the presence of background noise, and auditory performance is reduced due to the difficulty in isolating different sound sources [4,5]. Music listening is another task that is unfavorably impacted in CI users [5,6].

Music provides emotional stability and enhances life satisfaction in humans, and CI users consider it to be the second most important acoustic stimulus after speech [7,8]. Music has a wider dynamic range and frequency spectrum than speech [7,9], and the signal processing mechanisms of a CI can lead CI users to experience difficulty in enjoying the wide acoustic spectrum of music [10]. Previous studies have shown that after CI surgery, many recipients avoid listening to music due to a decrease in their music appreciation compared with the satisfaction experienced before hearing loss [8].

In recent years, the number of elderly individuals who have undergone CI surgery has increased as the elderly population with hearing loss has expanded. With aging, deterioration of the central auditory pathway (central presbycusis), as well as cochlear hair cell loss (peripheral presbycusis), progress [11,12]. In addition, the ability to perceive temporal cues is reduced in older adults compared to that in younger adults [13]. Therefore, music appreciation in elderly adults with a CI may differ from that in non-elderly adults. For example, in a previous study using simulated CI hearing in non-CI users with normal or near-normal hearing, the ability to recognize melody and timbre in older adults was poorer compared to that in younger adults [14]. However, little is known about the potential differences between non-elderly and elderly CI users.

The aims of the present study were to compare music appreciation between non-elderly and elderly CI recipients and to characterize both groups according to their music appreciation.

## 2. Materials and Methods

### 2.1. Subjects

The study population of this cross-sectional analysis included adults aged 18 years and older who underwent unilateral cochlear implantation from March 2009 to April 2021 and had used a CI for at least six months. Patients with hearing loss due to temporal bone trauma and/or meningitis, which might affect brain function, and those with cognitive impairments were excluded. The subjects were categorized into two groups based on age, with the non-elderly group comprising those aged 18–64 years and the elderly group composed of those aged 65 and older.

### 2.2. Demographic and Clinical Variables

Demographic information, preoperative auditory performance, intraoperative findings, types of speech processor used, postoperative auditory performance, postoperative functional residual hearing, hearing level in the non-implanted ear, and follow-up period were assessed to compare the basic and clinical characteristics of the two groups.

Auditory performance was evaluated using a disyllabic word test, sentence test, and category of auditory performance (CAP) score. Both disyllabic word and sentence tests were performed at a 65 dB sound pressure level (SPL) in a quiet environment, and the percentage of correct discrimination was determined. The Korean version of the Central Institute for the Deaf sentence was used without visual cues for the sentence test. CAP score was categorized into eight levels from 0 to 7 as described in a previous study [15]. To determine the hearing level in the non-implanted ear, pure-tone average was calculated using 500, 1000, 2000, and 3000 Hz, as recommended by the American Academy of Otolaryngology Committee on Hearing and Equilibrium [16].

### 2.3. Assessment of Music Appreciation

Music appreciation was evaluated before and after CI surgery using the Korean version of the Musical Background Questionnaire (K-MBQ), which is a validated questionnaire adapted from the Iowa MBQ [8] and translated into Korean [17]. The K-MBQ assesses formal musical training, music enjoyment, music quality, and music elements.

Formal musical training includes formal musical lessons and self-reported musical background. Formal music lessons were rated using questions in the following six categories: (1) musical instrument lessons, (2) singing lessons, (3) participation in an ensemble, (4) music lessons received in elementary school, (5) music lessons in middle school, and (6) music appreciation classes attended. Self-reported musical backgrounds were rated and quantified using the following five classifications: (1) no formal training, little knowledge about music, and little experience in listening to music (0 points); (2) no formal training or knowledge about music but informal listening experience (1 point); (3) self-taught musician who participates in musical activities (2 points); (4) some musical training, basic knowledge of musical elements, and participation in music classes or ensembles (3 points); and (5) several years of musical training, knowledge about music, and involvement in music groups (4 points).

Music enjoyment was assessed based on the duration of time spent listening to music in one week. Listening time was categorized as follows: 2 h or fewer, 3–5 h, 6–8 h, and ≥9 h. Music quality was scored using 6 10-point visual analog scales (VAS) with 14 bipolar adjective descriptions in quiet situations as follows: (1) unpleasant–pleasant (pleasantness), (2) mechanical–natural (naturalness), (3) fuzzy–clear (clarity), (4) does not sound like music–does sound like music (musicality), (5) complex–simple (non-complexity), and (6) difficult to follow–easy to follow (ease of following along).

Music elements were evaluated by the following six questions: (1) Can you hear the difference between singing and speaking?; (2) Are you able to differentiate between a male and a female vocalist?; (3) Are you able to follow the rhythm of a music piece?; (4) Are you able to recognize the melody of a music piece?; (5) Are you able to differentiate the instruments in a piece of music; and (6) Can you follow the lyrics of a song? Each question was scored on a scale from 1 (never) to 7 (always).

Preoperative and postoperative questionnaires were both completed after the implantation. CI recipients receiving mapping and/or speech therapy were evaluated using the K-MBQ under the supervision of their speech therapist. For those not receiving mapping and/or speech therapy, the K-MBQ was sent to them through the mail to be completed. Out of a total of 121 adult CI recipients, 49 (40.5%) (31 non-elderly and 18 elderly CI recipients) responded and completed the questionnaire.

### 2.4. Statistical Analysis

The Mann–Whitney U test was used to compare patient age, hearing loss duration, preoperative CAP and sentence scores, postoperative CAP and sentence scores, follow-up period, and musical background score between the two groups. Chi-square or Fisher’s exact tests were used to compare sex, implanted ear, formal musical training, and changes in music listening time and music genre between the two groups. Fisher’s exact test was performed when more than 20% of cells had expected frequencies < 5. Wilcoxon signed rank test was used to compare pre- and postoperative music quality and music element scores in each group.

Statistical Package for the Social Sciences (SPSS) version 22 software (SPSS Inc., Chicago, IL, USA) was used for statistical analyses. A *p*-value < 0.05 was considered statistically significant.

## 3. Results

### 3.1. Demographic and Clinical Characteristics of Participants

The average ages of the individuals in the non-elderly group (*n* = 31) and the elderly group (*n* = 18) were 47 (range: 25–63) and 73 (range: 66–80) years, respectively. There was no difference in terms of sex, implanted ear, duration of hearing loss, preoperative audiologic performances, type of speech processor used, postoperative audiologic performance, hearing level of the non-implanted ear, or follow-up period duration between the two groups (Table 1). The majority of CI users in both groups had received formal musical training. Neither the number of formal musical lesson categories nor the musical background level significantly differed between the two groups (*p* = 0.24 and *p* = 0.43, respectively) (Table 2).

### 3.2. Comparison of Music Enjoyment between the Non-Elderly and Elderly Groups

After CI surgery, a decrease in music listening time was observed in 55.6% (10 of 18) of the elderly group and in 25.8% (8 of 31) of the non-elderly group; the difference was statistically significant (*p* = 0.04). The genre of music listened to changed in 5.6% (1 of 18) of the elderly group and in 32.3% (10 of 31) of the non-elderly group; this difference was also statistically significant (*p* = 0.04) (Table 3).

### 3.3. Comparison of Music Quality between the Non-Elderly and Elderly Groups

Postoperatively, both groups exhibited a decrease in scores for all items assessing music quality. Among the items, significant changes between the preoperative and postoperative scores were observed for naturalness (8.5 vs. 3.5, *p* = 0.01), clarity (7.5 vs. 3.0, *p* = 0.01), similarity of sound to music (8.0 vs. 5.0, *p* = 0.02), non-complexity (8.5 vs. 4.0, *p* = 0.01), and ease of following along (7.0 vs. 4.0, *p* = 0.02) in the elderly group. The changes in the non-elderly group were not statistically significant (Figure 1).

### 3.4. Comparison of Music Element Scores between the Non-Elderly and Elderly Groups

All music element scores decreased after surgery in both the non-elderly and elderly groups. Among the items included in the musical element assessment, significant changes in the preoperative and postoperative scores for rhythm (6.0 vs. 4.0, *p* = 0.01), melody (6.0 vs. 4.0, *p* = 0.01), timbre (instruments differentiation) (6.5 vs. 4.0, *p* = 0.02), and lyrics (6.0 vs. 3.0, *p* = 0.01) were observed in the elderly group. The non-elderly group did not exhibit significant changes in scores for appreciation of music elements after CI surgery (Figure 2).

## 4. Discussion

Although many studies have been conducted on music appreciation in adult CI users, little is known about elderly CI users due to the fact that most studies do not distinguish between non-elderly and elderly adults. Therefore, we investigated whether music appreciation after CI surgery differs between non-elderly and elderly adults. The elderly group demonstrated significant changes in music appreciation in terms of both music quality and music elements. More specifically, they had significantly lower scores postoperatively than preoperatively in terms of naturalness, clarity, similarity of sounds to music, non-complexity, rhythm, melody, timbre, and lyrics. Conversely, the non-elderly group did not exhibit any significant changes in music appreciation after CI surgery, although a tendency toward decreased scores was observed. These findings suggest that elderly adults are susceptible to changes in music appreciation after CI surgery.

Through CI processing mechanisms, most information related to temporal fine structure is removed from the sound source, whereas temporal envelope information is preserved. Due to this preservation of temporal envelope information, CI users usually perceive rhythm to be similar to that of listeners with normal hearing [18,19,20,21,22]. Interestingly, we observed that the elderly group experienced a significant decreases in postoperative rhythm scores as compared with preoperative baseline values; no such changes were observed in the non-elderly group. This can be explained by the age-related decrease in the ability to perceive temporal envelope cues. Previous studies using electrophysiologic and psychoacoustic tests have shown that temporal envelope processing is reduced in older adults compared to that in young adults, regardless of their hearing level. Older listeners have been reported to exhibit less phase-locked activity in the auditory steady-state response and higher mean gap detection thresholds than younger listeners [23,24]. Older listeners also demonstrated reduced unmasking for normal speech at high frequencies compared to that in younger listeners [25].

Coarse resolution of the temporal fine structure mentioned above makes it difficult for CI users to perceive melody and timbre because spectral aspects of sound are relayed by the temporal fine structure. In the present study, this difficulty was observed in both groups after surgery, although it was more pronounced in the elderly group. This result can be attributed to the decreased ability to perceive temporal fine structure cues in elderly adults. A previous study that investigated the effects of age on temporal fine structures demonstrated that older adults exhibited poorer performance on a test of melodic pitch perception than younger adults, indicating a decrement in temporal fine structure processing associated with aging [13].

A significant decrease in music listening time, reflecting music enjoyment, was observed in the elderly group, which is consistent with the findings of previous studies. For example, a study of 35 CI users reported a positive correlation between enjoyment of music and younger age [26]. In another study, among the various preoperative factors, such as demographics, musical background, music listening habits, and speech outcomes, only age was correlated with music enjoyment, with the older CI listeners exhibiting lower music enjoyment scores than younger CI listeners [8]. Another study investigating optimal musical training programs also reported a negative correlation between age and music listening time [27].

Despite the dissatisfaction with music quality and the difficulty in understanding music elements, the elderly group did not change the genre of music they listened to after CI surgery. This could be explained by the fact that the group had a preferred music genre before undergoing surgery. The majority of individuals in the elderly group enjoyed ’trot’, a traditional Korean folk genre that has a slow rhythm and includes less complex instrumental and melodic compositions than those of other Korean pop genres. This uncomplicated structure might be less affected by signal processing of the CI. Similar results have been observed in studies of CI recipients whose main language was English and who listened to Western music; they preferred country and Western genres rather than classic, pop, or rock genres in which various musical instruments and complex melodic structures are employed [27,28].

This questionnaire-based study is subject to several limitations. First, a selection bias may exist, as the response rate was approximately 40.5%. Second, there may be a recall bias because the preoperative questionnaire was completed during the postoperative follow-up period. Third, the time of assessment of the questionnaire also varied from 6.3 to 123 months. However, most postlingual adult CI users reach a plateau in their audiologic performance within 6 months. The subjects in the present study were CI users for at least 6 months [3,29,30]. Therefore, the time point is not likely to have had a significant impact on our results. Fourthly, the preoperative scores of complexity, melody, timbre, and lyrics in the elderly group were higher than those in the non-elderly group. Although the there was no significant difference in formal musical lesson background between the two groups, the elderly group appears to have had slightly less experience with musical lessons, which might have led elderly CI users to rate the preoperative status generously, which may have caused overestimation of music appreciation before the implantation. Finally, no objective music tests were conducted, so a quantitative comparison with other studies is impossible. Further research involving detailed musical lesson evaluation and objective tests, such as the clinical assessment of music perception test [31], is needed to reinforce and validate our findings.

## 5. Conclusions

Our results suggest that elderly individuals are susceptible to changes in music appreciation following CI surgery. Although expectations of CI outcomes may vary from person to person, for elderly patients who rely heavily on music listening to overcome negative psychosocial changes that occur with aging, it is necessary to counsel them to set realistic expectations about music appreciation before surgery based on our results. In addition to surgical counselling, training programs to improve rhythm perception should be included in music training for elderly CI users during rehabilitation because a significant decrease in the perception of rhythm occurs relative to non-elderly CI users.

## Figures and Tables

**Figure 1 jcm-11-05029-f001:**
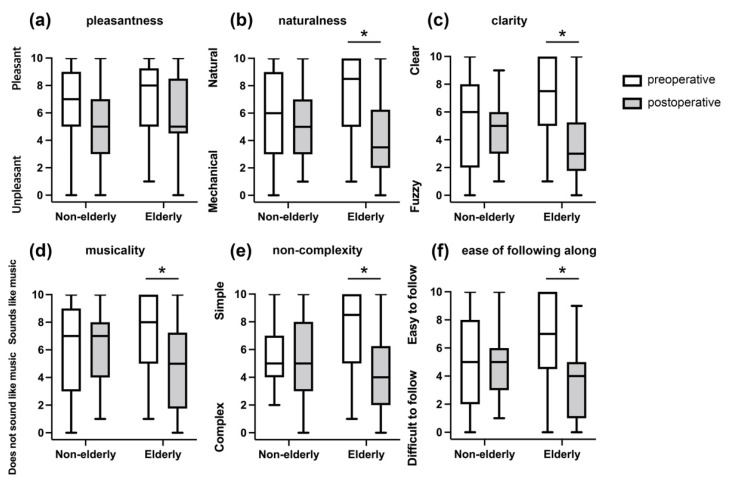
Comparison of perceived music quality between the non-elderly group and the elderly group before and after receiving a cochlear implant. (**a**) Pleasantness; (**b**) naturalness; (**c**) clarity; (**d**) musicality; (**e**) non-complexity; (**f**) ease of following along. The elderly group exhibits significant decreases in the postoperative scores for naturalness, clarity, similarity of sounds to music, non-complexity, and ease of following along compared with their preoperative scores. No significant change is observed for any score in the non-elderly group. * *p* < 0.05.

**Figure 2 jcm-11-05029-f002:**
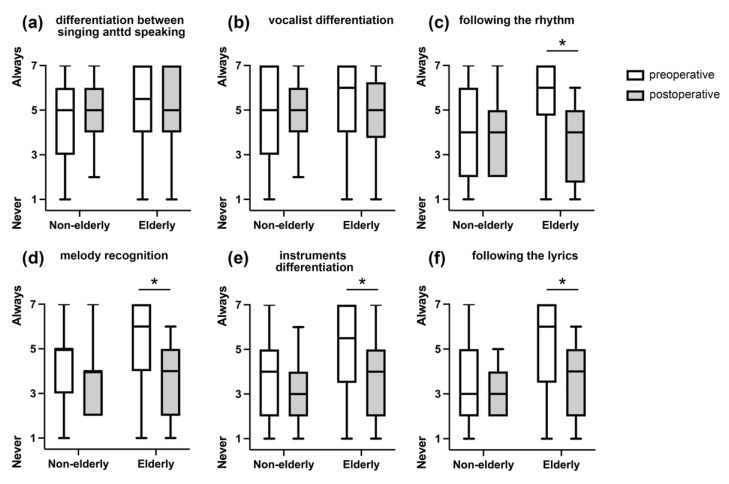
Comparison of music elements between the non-elderly group and the elderly group before and after receiving a cochlear implant. (**a**) Ability to perceive the difference between singing and speaking; (**b**) ability to differentiate between a male and a female vocalist; (**c**) ability to follow the rhythm; (**d**) ability to recognize the melody; (**e**) ability to differentiate the instruments; (**f**) ability to follow the lyrics. The elderly group shows significant declines in postoperative scores for rhythm, melody, timbre, and lyrics compared with their preoperative scores. No significant change is observed for any factor in the non-elderly group. * *p* < 0.05.

**Table 1 jcm-11-05029-t001:** Demographic and clinical characteristics of participants.

	Non-Elderly Group (*n* = 31)	Elderly Group (*n* = 18)	*p* Value
Mean age (SEM) (range), y	47 (2.1) (25–63)	73 (0.9) (66–80)	<0.01 *
Sex, *n* (%)			0.81
Male	11 (35.5)	7 (38.9)	
Female	20 (64.5)	11 (61.1)	
Implanted ear, *n* (%)			0.67
Right	14 (45.2)	7 (38.9)	
Left	17 (54.8)	11 (61.1)	
Mean HL duration (SEM) (range), y	9 (1.7) (0.5–30)	6 (1.5) (0.5–25)	0.56
Mean preoperative CAP score (SEM) (range)	1.2 (0.4) (0–4)	1.7 (0.5) (0–4)	0.41
Mean preoperative WDS (SEM) (range), %	7 (2.8) (0–43)	13 (4.2) (0–42)	0.12
Mean preoperative sentence score (range), %	8 (3.2) (0–48)	16 (4.8) (0–46)	0.10
Speech processor, *n* (%)			0.31
N5/N6/N7	15 (48.4)	6 (33.3)	
OPUS2/Sonnet	16 (51.6)	12 (66.7)	
Mean postoperative CAP score (SEM) (range)	5.9 (0.2) (4–7)	5.6 (0.2) (5–6)	0.60
Mean postoperative WDS (SEM) (range), %	66 (4.6) (30–100)	70 (5.5) (60–77)	0.67
Mean postoperative sentence score (SEM) (range), %	75 (2.3) (39–100)	84 (2.6) (67–100)	0.34
Mean PTA in non-implanted ear (SEM) (range), dB HL	94 (5.3) (85–120)	89 (6.7) (85–120)	0.51
Mean follow-up period (SEM) (range), y	4.5 (0.6) (0.5–10.3)	5.0 (0.7) (0.5–9.8)	0.90

* *p* < 0.05 indicates a statistically significant difference between the two groups for a given parameter. SEM, standard error of the mean; HL, hearing loss; CAP, category of auditory performance; WDS, word discrimination score; PTA, pure-tone average; dB HL, decibel hearing level.

**Table 2 jcm-11-05029-t002:** Comparison of formal musical training and musical background between the non-elderly and elderly groups.

	Non-Elderly Group (*n* = 31)	Elderly Group (*n* = 18)	*p* Value
Formal musical lesson, *n* (%)			0.08
No	2 (6.5)	5 (27.8)	
Yes	29 (93.5)	13 (72.2)	
Number of formal musical lesson category, *n* (%) ^a^			0.24
One category	10 (34.5)	7 (53.8)	
Two or more categories	19 (65.5)	6 (46.2)	
Mean musical background level (range)	3.2 (1–5)	3.0 (1–5)	0.43

^a^ 42 patients were analyzed. *p* < 0.05 indicates a statistically significant difference between the two groups for a given parameter.

**Table 3 jcm-11-05029-t003:** Comparison of changes in music listening time and music genre between the non-elderly and elderly groups.

	Non-Elderly Group (*n* = 31)	Elderly Group (*n* = 18)	*p* Value
Decrease in music listening time, *n* (%)			0.04 *
Yes	8 (25.8)	10 (55.6)	
No	23 (74.2)	8 (44.4)	
Change in music genre, *n* (%)			0.04 *
Yes	10 (32.3)	1 (5.6)	
No	21 (67.7)	17 (94.4)	

* *p* < 0.05 indicates a statistically significant difference between the two groups for a given parameter.

## Data Availability

Data are available upon reasonable request from the corresponding author.

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
