# Peer review of "Susceptibility to Postoperative Changes in Music Appreciation in Elderly Cochlear Implant Recipients"

_jcm, 2022, doi:10.3390/jcm11175029_

Round 1
Reviewer 1 Report
The study investigated the change in music appreciation after a cochlear implantation in an elderly group (>65 years) and a younger group (<65 years). A group of 49 participants were investigated via questionnaires. The elderly group demonstrated a decrease in listening time and decreases in different scores for perceived music quality and music elements. For the non-elderly group such significant differences between the pre- and post-operative state were not observed.
The study is very interesting and important for the estimation of the quality of life regarding music appreciation in elderly CI user. However, I have some comments. My main concern is that the results are mainly due to differences in the estimation of the preoperative condition by participants in the two groups.
Major points:
(1) The participants received a unilateral cochlear implant. Information about the contralateral ear should be provided because it is essential for the interpretation of the data. Was hearing still present in the non-implanted ear? Was there a difference in the residual hearing between the groups?
(2) In the elderly group, the differences between pre- and postoperative state seems to be mainly due to higher values in the preoperative state in the elderly compared to the non-elderly group (e.g. for complexity, melody recognition, timbre, lyrics). Were the differences in the preoperative states tested between groups? In the Material and Methods section, it should be clearly stated that the preoperative and postoperative questionnaires are both filled in after the implantation as stated in the Discussion about a possible recall bias. Could it be that all the observed effects are due to an overestimation of the music appreciation before the implantation in the elderly group? Might this be related to the slightly lower formal musical training in the elderly group?
(3) How was the border between the non-elderly and elderly group defined? Is there any reason for taking 65 years as border? As there is no apparent age gap between the groups correlation between age and the different scores might be more appropriate.
(4) A large part of the conclusion is a repetition of results. How could the results concretely used for surgical counselling and music training programs?
(5) It would be good if more recent literature on listening to music in CI users were included in the introduction and discussion.
Minor points:
(6) Page 1 line 38: Citations for the sentence about music listening should be provided.
(7) How were the category of auditory performance, sentence and word discrimination scores determined? The procedure or citations should be provided.
(8) Were corrections for multiples tests performed?
(9) Table 1 should be checked because there are some inconsistencies, e.g. mean age total 25-85 but for elderly 66-80; mean HL duration total cannot be 10.2 when non-elderly 9.2 and elderly 6.1
(10)Page 4 line 156: one “number” should be deleted
(11)Page 5 lines 159/160: In this sentence, the values of the elderly and non-elderly group are reversed.
(12)Figure 1: Headlines for each diagram would be helpful. The single letters are not picked up anywhere and therefore are not necessary or they should be picked up somewhere, e.g., in the caption of the figure.
Author Response
Major points:
- The participants received a unilateral cochlear implant. Information about the contralateral ear should be provided because it is essential for the interpretation of the data. Was hearing still present in the non-implanted ear? Was there a difference in the residual hearing between the groups?
Response: The pure-tone average of non-implanted ear in the non-elderly group and the elderly group was 94 (range, 85–120) and 89 (range, 85–120) dB HL, respectively, which was not significantly different (P=0.51). In response to this comment, we added the above information to the Results (page 3, line 140) and Table 1.
We also added the following sentence to the Materials and Methods with a related citation:
“For hearing level in non-implanted ear, pure-tone average was calculated using 500, 1000, 2000, and 3000 Hz, as recommended by the American Academy of Otolaryngology Committee on Hearing and Equilibrium [16] (page 2, lines 80–83).
2-1. In the elderly group, the differences between pre- and postoperative state seems to be mainly due to higher values in the preoperative state in the elderly compared to the non-elderly group (e.g. for complexity, melody recognition, timbre, lyrics). Were the differences in the preoperative states tested between groups?
Response: As you pointed out, the preoperative scores of non-complexity, melody, timbre, and lyrics in the elderly group was significantly higher than those in the non-elderly group. However, in the present study, we focused on the changes of those variables in each group after CI surgery. Therefore, we compared postoperative values with preoperative ones using the statistical method for paired sample within each group (i.e. Wilcoxon signed rank test) as described in Materials and Methods section.
2-2 In the Material and Methods section, it should be clearly stated that the preoperative and postoperative questionnaires are both filled in after the implantation as stated in the Discussion about a possible recall bias.
Response: We thank the reviewer for raising this point. We added the following statement to the Materials and Methods section:
Preoperative and postoperative questionnaires were both filled in after the implantation (page 3, lines 115–116).
2-3 Could it be that all the observed effects are due to an overestimation of the music appreciation before the implantation in the elderly group? Might this be related to the slightly lower formal musical training in the elderly group?
Response: We agree with your opinion and added the following sentences to the Discussion section as our limitation:
“Fourthly, the preoperative scores of non-complexity, melody, timbre, and lyrics in the elderly group was higher than those in the non-elderly group. Although the there was no significant difference in formal musical lesson background between the two groups, the elderly group appears to have slightly lower experiences of musical lesson. This less experience in musical lessons might lead for the elderly CI users to rate the preoperative status generously, which can cause overestimation of music appreciation before the implantation” (page 8, lines 263-268).
- How was the border between the non-elderly and elderly group defined? Is there any reason for taking 65 years as border? As there is no apparent age gap between the groups correlation between age and the different scores might be more appropriate.
Response: Several previous studies on CI in the elderly population divided the elderly and the non-elderly individuals by the age of 65 years as demonstrated in the following articles:
- Mosnier, I.; Bebear, J.P.; Marx, M.; Fraysse, B.; Truy, E.; Lina-Granade, G.; Mondain, M.; Sterkers-Artieres, F.; Bordure, P.; Robier, A.; et al. Improvement of cognitive function after cochlear implantation in elderly patients. JAMA Otolaryngol Head Neck Surg 2015, 141, 442-450, doi:10.1001/jamaoto.2015.129.
- Leung, J.; Wang, N.Y.; Yeagle, J.D.; Chinnici, J.; Bowditch, S.; Francis, H.W.; Niparko, J.K. Predictive models for cochlear implantation in elderly candidates. Arch Otolaryngol Head Neck Surg 2005, 131, 1049-1054, doi:10.1001/archotol.131.12.1049.
- Migirov, L.; Taitelbaum-Swead, R.; Drendel, M.; Hildesheimer, M.; Kronenberg, J. Cochlear implantation in elderly patients: surgical and audiological outcome. Gerontology 2010, 56, 123-128, doi:10.1159/000235864.
- Sonnet, M.H.; Montaut-Verient, B.; Niemier, J.Y.; Hoen, M.; Ribeyre, L.; Parietti-Winkler, C. Cognitive Abilities and Quality of Life After Cochlear Implantation in the Elderly. Otol Neurotol 2017, 38, e296-e301, doi:10.1097/MAO.0000000000001503.
In addition, Korea, the country where the present study was performed, defines elderly as those aged 65 years and older, taking into account the age for retirement from work. Therefore, we set 65 years as the age criteria for the elderly.
- A large part of the conclusion is a repetition of results. How could the results concretely used for surgical counselling and music training programs?
Response: We thank the reviewers for raising this up. We have removed the following redundant statements from the Conclusion section
- Elderly individuals are more likely than non-elderly adults to exhibit noticeable changes in music appreciation, in terms of both music quality and music elements, after undergoing CI surgery. They are likely to experience significant postoperative decline in elements related to the perception of music naturalness and clarity, similarity of sounds to music, non-complexity of music, and ease of following along. They are also likely to have significant postoperative decline in the perception of rhythm, melody, timbre, and lyrics. Collectively,
In addition, we have revised our conclusion statement with adding implications of our results on surgical counselling and music training program as follow:
- Our results suggest that elderly individuals are susceptible to changes in music appreciation following CI surgery. Although expectations of CI outcomes may vary from person to person, for elderly patients who rely heavily on music listening to overcome negative psychosocial changes that occurs with aging, it is necessary to consult before surgery to set realistic expectations about music appreciation based on our results. In addition to the surgical counselling, training programs to improve rhythm perception should be included in music training for elderly CI users during rehabilitation because a significant decrease in the perception of rhythm occurs contrary to non-elderly CI users (pages 8–9, lines 274–281).
- It would be good if more recent literature on listening to music in CI users were included in the introduction and discussion.
Response: We agree with the reviewer’s advice and have therefore added the following references to relevant sentences in the Introduction and Discussion sections.
- Jiam, N.T.; Caldwell, M.T.; Limb, C.J. What Does Music Sound Like for a Cochlear Implant User? Otol Neurotol 2017, 38, e240-e247, doi:10.1097/MAO.0000000000001448.
- Bruns, L.; Murbe, D.; Hahne, A. Understanding music with cochlear implants. Sci Rep 2016, 6, 32026, doi:10.1038/srep32026.
- Jiam, N.T.; Limb, C.J. Rhythm processing in cochlear implant-mediated music perception. Ann N Y Acad Sci 2019, 1453, 22-28, doi:10.1111/nyas.14130.
Minor points:
- Page 1 line 38: Citations for the sentence about music listening should be provided.
Response: We thank the reviewer for their advice. We have added the following citations on music listening in the Introduction section:
“Music listening is another task that is unfavorably impacted in CI users [5,6]” (page 1, line 39).
- Gfeller, K.; Jiang, D.; Oleson, J.J.; Driscoll, V.; Olszewski, C.; Knutson, J.F.; Turner, C.; Gantz, B. The effects of musical and linguistic components in recognition of real-world musical excerpts by cochlear implant recipients and normal-hearing adults. J Music Ther 2012, 49, 68-101, doi:10.1093/jmt/49.1.68.
- Gfeller, K.E.; Olszewski, C.; Turner, C.; Gantz, B.; Oleson, J. Music perception with cochlear implants and residual hearing. Audiol Neurootol 2006, 11 Suppl 1, 12-15, doi:10.1159/000095608.
- How were the category of auditory performance, sentence and word discrimination scores determined? The procedure or citations should be provided.
Response: We thank the reviewer for raising this up. The auditory performance, sentence discrimination, and word discrimination scores were evaluated using the category of auditory performance (CAP) score, sentence test, and disyllabic word test, respectively. We have added the following statement to the Materials and Methods with a related citation:
Audiologic performances were evaluated using disyllabic word test, sentence test, and category of auditory performance (CAP) score. Both disyllabic word and sentence tests were performed at 65 dB sound pressure level (SPL) in a quiet environment, and the percentage of correct discrimination were determined. Korean version of the Central Institute for the Deaf sentence was used without visual cues for the sentence test. CAP score was categorized into eight levels from 0 to 7 as described in a previous study [15] (page 2, lines 75–80).
- Archbold, S.; Lutman, M.E.; Marshall, D.H. Categories of Auditory Performance. Ann Otol Rhinol Laryngol Suppl 1995, 166, 312-314.
- Were corrections for multiples tests performed?
Response: We focused on change of those variables within each group after CI surgery. Therefore, we compared postoperative values with preoperative ones using the statistical method for paired sample within each group (i.e. Wilcoxon signed rank test) as described in Materials and Methods section. Thus, multiples tests with post-hoc comparison was unnecessary.
- Table 1 should be checked because there are some inconsistencies, e.g. mean age total 25-85 but for elderly 66-80; mean HL duration total cannot be 10.2 when non-elderly 9.2 and elderly 6.1
Response: We thank you for raising this up and apologize for the inconsistency. We have corrected the mean HL duration from “10.2” to “7.8” in Table 1. However, the second reviewer suggested to remove the first column showing total population from the table because it did not provide any further information. Therefore, we removed the information of total subjects from Table 1.
- Page 4 line 156: one “number” should be deleted
Response: In response to this comment, we have removed it and made the following changes in the footnote of Table 2:
“The number of patients analyzed is 42”
- Page 5 lines 159/160: In this sentence, the values of the elderly and non-elderly group are reversed.
Response: We agree with you and corrected the values for music listening time in the elderly and non-elderly groups as follows:
“…a decrease in music listening time was observed in 55.6% (10 of 18) of the elderly group and in 25.8% (8 of 31) of the non-elderly group…” (page 5, lines 161–162).
- Figure 1: Headlines for each diagram would be helpful. The single letters are not picked up anywhere and therefore are not necessary or they should be picked up somewhere, e.g., in the caption of the figure.
Response: In response to this comment, we have added headlines for each diagram and added the single letter legends in the caption of Figure 1.

Reviewer 2 Report
This manuscript treats the changes on music appreciation in cochlear implant recipients comparing two groups – elderly and non-elderly patients. The manuscript also provides insights about postoperative outcomes compared to the preoperative status after unilateral CI. The central point is that musical appreciation decreases after cochlear implantation in general but elderly patients could be more susceptible to exhibit a lower music appreciation than non-elderly patients. Below, I provide some questions and my detailed review report to the authors:
Introduction section
Line 37: I think that the term “speech performance” is not adequate and must be modified. Use the term “hearing performance” or “auditory performance” instead of speech performance. Here, we talk about the speech recognition of the language not the language production.
Line 47: the term “with the advancement of age” is not clear. Use “with aging” instead of with the advancement of the age”.
The aim of the study should be changed a little for better clarity. The main goal of the study was to “compare” two populations non-elderly and elderly CI recipients and off course to characterize both groups according to their music appreciation. Change the objective in this direction.
Material and methods section
In the subjects sub-section was not stated the type of study. As stated by authors on the discussion section, the music appreciation was assessed on the postoperative period. Was the study a retrospective? Cross-sectional? This should be stated on the M&M section
Line 61: There were included patients who underwent CI surgery between 2009 to 2021, and were CI users for at least 6 months. When was the music appreciation assessed? For example, the assessment of the music appreciation of a patient implanted in 2009 could be different if we assess this patient in 2010 (assessment 1 year after surgery) or in 2021 (assessment 12 years after surgery).
Line 63: There were excluded patients with pathologies associated with a probable alteration of central nervous system compromise. However, the authors stated that there were excluded patients with cognitive impairment. How was the cognitive impairment detected or assessed to exclude a patient? On the preoperative? On the postoperative period? Which was the instrument used? With a MMT?
The sub-sections of the M&M section should have the same order of the result section. At first the “Clinical and demographic variables”, then assessment of music appreciation” and then an independent sub-section “statistical analysis”
Line 102 and 103 – hearing and speech performances: what is the difference between hearing and speech performances? Was the oral production assessed in this study? speech performance = language production. I think that hearing performance comprise the speech recognition and music appreciation. As the introduction speech performance is not the correct term. Explain and write the correct term in the text please.
Line 106 – (CAP): All abbreviations should be explained when it appears for the first time. How is it assessed the pre or postoperative CAP? Explain in the text
Line 106: what is the meaning of "sentence score"? How was it assessed? Does this term suggests that language production was evaluated?
In the results section (table 1), we observe that word discrimination score were evaluated, however on M&, there was neither observed a description of WDS nor how the WDS was evaluated. Were there used monosyllabic, disyllabic words?
Line 107: Chi-square test or Fisher tests were used in the analysis of contingency tables. However, it is not clear when the authors used Chi square or Fisher tests.
Results section:
Line 118: Use the whole number for the age (47 years) instead of 46.6 years. The same rule when age is expressed in numbers.
Line 123: the sentence training number of music training category is not clear. Explain and rephrase
Line 121-125: the authors stated that both groups had received some musical training but neither training… nor… significantly differed. However, we observe 3 p values between parentheses. Rephrase, the sentence is not clear.
Table 1: In this study, we compare two groups – elderly and non–elderly. However, in table 1, we observe a column of the whole population – total. It is not correct to put together two different groups. I don’t understand the interest of the column Total.
Table 1: For easy understand, authors must avoid to use a lot of decimals. For example, age should be expressed as 47 and 73, mean hearing loss 9 and 6, mean preop CAP 1.2 and 1.7, for the mean WDS avoid 66.0 (30.0–100) etc. The message is the same if we use less decimals and for the reader could be easy to understand. In the case of p value, they had to be rounded to two decimal places. The same for the other tables if applicable.
Table 1: When we read a result, we need to have an idea how the population is. This information is given by four parameters – the mean, the SD, min and max. In this case, the size of the population is different on the elderly and non-elderly groups. Consequently, it should be added the SEM to have an idea of the statistical dispersion to compare both groups in each case.
Table 1: When was the musical appreciation assessed? Was the time of the assessment the same for all patients? if not you have to state this in the table as a result
Line 150: We already know that a p value of 0.05 was considered as significant. Maybe, authors might use an asterisk * p<0.05, ** p<0.01 instead of a as appropriate.
Line 151: Order the abbreviations according to the order of appearance of the term in the table.
Table 2: The first column is not clear because of the alignment. The main titles: formal music training, number of musical training category and mean musical background level should be aligned to the left and the subtitles should be aligned all to the same level but a little more to the right. Please format all the tables in the same way.
Table 2: As stated on the M&M, all patients filled the questionnaire. Why was the number of patients different (n=42), only for the number of musical training?
Figures 1 and 2: Authors present the analysis of the perception of music quality and the music elements factor by factor. In each case, we have a numeric variable (e.g. naturalness – unpleasant to pleasant), and two categorical variables 1. Elderly /non-elderly and 2. Preoperative/postoperative. Consequently, we need to answer these questions:
1. Were the preoperative values different in both groups? Sometimes, we observe that the values of the elderly group were better than non-elderly group
2. Were the postoperative values different in both groups?
The goal of the study was to compare elderly and non-elderly CI users. So, we need to answer this question according to the objective. For example, the naturalness, the x axis could be preoperative and postoperative instead of non-elderly and elderly. In the case of the color of the boxplots, it could be non-elderly and elderly instead of preimplant and postimplant.
3. Were the preoperative and the postoperative values different inside each group (non-elderly or elderly)? Was the deterioration of the musical appreciation more marked in the elderly than the non-elderly group?
Sometimes I observe that the difference between the value pre and postoperative in the non-elderly group decrease less than the values observed in the elderly group (e.g. easy of following).
Figure 1: The scale (y-axis) of all figures are 1, 3, 5, 7 with the exception of (f). Please use the same format for all figures
Line 179 -180: Put the corresponding letter of the figure in the legend (e.g. postoperative scores for (a) naturalness, (b) clarity…
In figure 1, the colors are expressed as preimplant and postimplant. In the legend, authors write postoperative and preoperative, write the same term in both.
Lines 198-199: This is not necessary to write (c), (d) or (e) twice.
Both figures need to have the same format. In figure 1, we observe only (a), (b)… however, in figure 2 we observe (a) and a description.
In the legend of the figure, authors have to state the meaning of the asterisk * p<0.05
Discussion section:
Concerning the hearing performance, we observe in the table 1 that the postoperative WDS was similar in both groups. Was the music appreciation similar in both groups in the postoperative period? was it different? Was there a relationship between the speech recognition scores (objective measure) and the music appreciation (subjective measure)?
Line 249-250: Avoid the verb to be “instrumental and melodic composition are” use another verb instead of "are"
Line 261-262: Authors recommend to perform objective tests to assess musical appreciation, however there was used the reference 24 at the end of the sentence. I think that this reference could be removed.
Conclusion section:
Line 264: Avoid the use of the verb to be, use another verb such as suggest instead of are.
At the end of the conclusion, add a little phrase concerning the perspectives or recommendations of this work to be studied in future trials.
Line 279: remove the work between parenthesis (research)
The text in the funding paragraph is repeated in the acknowledgement. Use this text either in the funding or in the acknowledgements.
Author Response
Introduction section
- Line 37: I think that the term “speech performance” is not adequate and must be modified. Use the term “hearing performance” or “auditory performance” instead of speech performance. Here, we talk about the speech recognition of the language not the language production.
Response: We agree with the reviewer and therefore changed “speech performance” to “auditory performance” in the Introduction section (page 1, line 37).
- Line 47: the term “with the advancement of age” is not clear. Use “with aging” instead of with the advancement of the age”.
Response: We agree with the reviewer and therefore changed “with the advancement of age” to “with aging” in the Introduction section (page 2, line 48).
- The aim of the study should be changed a little for better clarity. The main goal of the study was to “compare” two populations non-elderly and elderly CI recipients and off course to characterize both groups according to their music appreciation. Change the objective in this direction.
Response: We thank the reviewer for highlighting the ambiguity. We have revised the aim of the study to address this as follows:
“The aims of the present study were to compare music appreciation between the two populations, non-elderly and elderly CI recipients, and to characterize both groups according to their music appreciation.” (page 2, lines 57–59).
Material and methods section
- In the subjects sub-section was not stated the type of study. As stated by authors on the discussion section, the music appreciation was assessed on the postoperative period. Was the study a retrospective? Cross-sectional? This should be stated on the M&M section
Response: The present study is a cross-sectional study. In response to this comment, we specified the study type in the Materials and Methods section as follows:
“The study population of this cross-sectional analysis included adults aged 18 years and older…” (page 2, line 62).
- Line 61: There were included patients who underwent CI surgery between 2009 to 2021, and were CI users for at least 6 months. When was the music appreciation assessed? For example, the assessment of the music appreciation of a patient implanted in 2009 could be different if we assess this patient in 2010 (assessment 1 year after surgery) or in 2021 (assessment 12 years after surgery).
Response: The time of assessment of the questionnaire varied from 6.3 to 123 months. However, most postlingual adult CI users reach a plateau in their audiologic performance within 6 months. The subjects in the present study were CI users for at least 6 months. Therefore, the time point is not likely to have a significant impact on our results. In response to this comment, we added the following statement and citations to our study limitations in the Discussion section:
Third, the time of assessment of the questionnaire also varied from 6.3 to 123 months. However, most postlingual adult CI users reach a plateau in their audiologic performance within 6 months. The subjects in the present study were CI users for at least 6 months [3,29,30]. Therefore, the time point is not likely to have a significant impact on our results. (page 8, lines 259–263)
- Mosnier, I.; Bebear, J.P.; Marx, M.; Fraysse, B.; Truy, E.; Lina-Granade, G.; Mondain, M.; Sterkers-Artieres, F.; Bordure, P.; Robier, A.; et al. Improvement of cognitive function after cochlear implantation in elderly patients. JAMA Otolaryngol Head Neck Surg 2015, 141, 442-450, doi:10.1001/jamaoto.2015.129.
- Knopke, S.; Haussler, S.; Grabel, S.; Wetterauer, D.; Ketterer, M.; Fluger, A.; Szczepek, A.J.; Olze, H. Age-Dependent Psychological Factors Influencing the Outcome of Cochlear Implantation in Elderly Patients. Otol Neurotol 2019, 40, e441-e453, doi:10.1097/MAO.0000000000002179.
- Vermeire, K.; Brokx, J.P.; Wuyts, F.L.; Cochet, E.; Hofkens, A.; Van de Heyning, P.H. Quality-of-life benefit from cochlear implantation in the elderly. Otol Neurotol 2005, 26, 188-195, doi:10.1097/00129492-200503000-00010.
- Line 63: There were excluded patients with pathologies associated with a probable alteration of central nervous system compromise. However, the authors stated that there were excluded patients with cognitive impairment. How was the cognitive impairment detected or assessed to exclude a patient? On the preoperative? On the postoperative period? Which was the instrument used? With a MMT?
Response: Before CI surgery, we routinely evaluate cognitive function using the Mini-Mental State Evaluation (MMSE) in all CI candidates. In addition, the MMSE is performed again during follow-up to assess the CI recipient’s cognitive function if necessary. The MMSE scores of the patients in the present study were all within the normal range.
- The sub-sections of the M&M section should have the same order of the result section. At first the “Clinical and demographic variables”, then assessment of music appreciation” and then an independent sub-section “statistical analysis”
Response: We thank the reviewer for highlighting this inconsistency. We changed the order of subsections of the Material and Methods to the following order:
2.2 Demographic and clinical variables; 2.3 Assessment of music appreciation; 2.4 Statistical analysis (lines 70, 85, 122).
- Line 102 and 103 – hearing and speech performances: what is the difference between hearing and speech performances? Was the oral production assessed in this study? speech performance = language production. I think that hearing performance comprise the speech recognition and music appreciation. As the introduction speech performance is not the correct term. Explain and write the correct term in the text please.
Response: We thank the reviewer for raising this point. We did not assess the oral production. In this study, the speech performance meant the ability of speech recognition. In response to this comment, we changed “preoperative hearing and speech performance” and “postoperative hearing and speech performance” to “preoperative auditory performance” and “postoperative auditory performance”, respectively (lines 71–72). We further described the method for evaluating the auditory performance in the Materials and Methods section as follows:
Auditory performances were evaluated using disyllabic word test, sentence test, and category of auditory performance (CAP) score. Both disyllabic word and sentence tests were performed at 65 dB sound pressure level (SPL) in a quiet environment, and the percentage of correct discrimination were determined. The Korean version of the Central Institute for the Deaf sentence was used without visual cues for the sentence test. CAP score was categorized into eight levels from 0 to 7 as described in a previous study [15]. For hearing level in non-implanted ear, pure-tone average was calculated using 500, 1000, 2000, and 3000 Hz, as recommended by the American Academy of Otolaryngology Committee on Hearing and Equilibrium [16] (page 2, lines 75–80).
9 and 10. Line 106 – (CAP): All abbreviations should be explained when it appears for the first time. How is it assessed the pre or postoperative CAP? Explain in the text
Line 106: what is the meaning of "sentence score"? How was it assessed? Does this term suggests that language production was evaluated?
Response: We thank the reviewer for their advice. We have included the abbreviation for CAP and added the following statements to the Materials and Methods section with relevant citations: “Auditory performances were evaluated using disyllabic word test, sentence test, and category of auditory performance (CAP) score. Both disyllabic word and sentence tests were performed at 65 dB sound pressure level (SPL) in a quiet environment, and the percentage of correct discrimination were determined. The Korean version of the Central Institute for the Deaf sentence was used without visual cues for the sentence test. CAP score was categorized into eight levels from 0 to 7 as described in a previous study [15]” (page 2, lines 75–80).
- Archbold, S.; Lutman, M.E.; Marshall, D.H. Categories of Auditory Performance. Ann Otol Rhinol Laryngol Suppl 1995, 166, 312-314.
- In the results section (table 1), we observe that word discrimination score were evaluated, however on M&, there was neither observed a description of WDS nor how the WDS was evaluated. Were there used monosyllabic, disyllabic words?
Response: We thank the reviewer for raising this point. We have used disyllabic words for the word test. In response to this comment, we have specified the tests used in the Materials and Methods section (lines 75-78).
- Line 107: Chi-square test or Fisher tests were used in the analysis of contingency tables. However, it is not clear when the authors used Chi square or Fisher tests.
Response: Fisher’s exact tests was applied when more than 20% of cells have expected frequencies <5. In response to this comment, we added the following sentence to the Materials and Methods section:
Fisher’s exact test was performed when more than 20% of cells have expected frequencies < 5 (page 3, lines 127–128).
Results section:
- Line 118: Use the whole number for the age (47 years) instead of 46.6 years. The same rule when age is expressed in numbers.
Response: We agree with the reviewer’s advice and changed the presentation of age to whole numbers in the Results section (page 3, line 137) and Table 1 (page 4, line 151).
- Line 123: the sentence training number of music training category is not clear. Explain and rephrase
Response: The training number of music training category means the number of formal musical lesson categories that the patients participated in as their musical background. In response to this comment, we changed “training number of music training category” to “the number of formal musical lesson categories” in the Results (page 3, line 142).
- Line 121-125: the authors stated that both groups had received some musical training but neither training… nor… significantly differed. However, we observe 3 p values between parentheses. Rephrase, the sentence is not clear.
Response: We thank the reviewer for highlighting this point. To improve the clarity, we revised the statement to:
“The majority of CI users in both groups had received formal music training. Neither the number of formal musical lesson category, nor the musical background level significantly differed between the two groups (P=0.24, and P=0.43, respectively) (page 3, lines 141–143).
- Table 1: In this study, we compare two groups – elderly and non–elderly. However, in table 1, we observe a column of the whole population – total. It is not correct to put together two different groups. I don’t understand the interest of the column Total.
Response: We thank the reviewer for raising this point. We intended to show the characteristics of the total population because the number of patients in each group was different. In response to this comment, we removed the column from Table 1.
- Table 1: For easy understand, authors must avoid to use a lot of decimals. For example, age should be expressed as 47 and 73, mean hearing loss 9 and 6, mean preop CAP 1.2 and 1.7, for the mean WDS avoid 66.0 (30.0–100) etc. The message is the same if we use less decimals and for the reader could be easy to understand. In the case of p value, they had to be rounded to two decimal places. The same for the other tables if applicable.
Response: We agree with the reviewer’s advice and changed the values accordingly in the text (lines 163, 165, 174-176, 189-191) as well as in Tables 1–3.
- Table 1: When we read a result, we need to have an idea how the population is. This information is given by four parameters – the mean, the SD, min and max. In this case, the size of the population is different on the elderly and non-elderly groups. Consequently, it should be added the SEM to have an idea of the statistical dispersion to compare both groups in each case.
Response: We agree with the reviewer’s advice and added the SEM to Table 1.
- Table 1: When was the musical appreciation assessed? Was the time of the assessment the same for all patients? if not you have to state this in the table as a result.
Response: We thank the reviewer for raising this point. The time of the assessment was not the same for all patients. We added the following statement in the Discussion section, as part of our limitations:
“… the time of assessment of the questionnaire also varied from 6.3 to 123 months. However, most postlingual adult CI users reach a plateau in their audiologic performance within 6 months [3,29,30]. The subjects in the present study were CI users for at least 6 months. Therefore, the time point is not likely to have a significant impact on our results.” (lines 259-262).
- Mosnier, I.; Bebear, J.P.; Marx, M.; Fraysse, B.; Truy, E.; Lina-Granade, G.; Mondain, M.; Sterkers-Artieres, F.; Bordure, P.; Robier, A.; et al. Improvement of cognitive function after cochlear implantation in elderly patients. JAMA Otolaryngol Head Neck Surg 2015, 141, 442-450, doi:10.1001/jamaoto.2015.129.
- Knopke, S.; Haussler, S.; Grabel, S.; Wetterauer, D.; Ketterer, M.; Fluger, A.; Szczepek, A.J.; Olze, H. Age-Dependent Psychological Factors Influencing the Outcome of Cochlear Implantation in Elderly Patients. Otol Neurotol 2019, 40, e441-e453, doi:10.1097/MAO.0000000000002179.
- Vermeire, K.; Brokx, J.P.; Wuyts, F.L.; Cochet, E.; Hofkens, A.; Van de Heyning, P.H. Quality-of-life benefit from cochlear implantation in the elderly. Otol Neurotol 2005, 26, 188-195, doi:10.1097/00129492-200503000-00010.
- Line 150: We already know that a p value of 0.05 was considered as significant. Maybe, authors might use an asterisk * p<0.05, ** p<0.01 instead of a as appropriate.
Response: We thank the reviewer for raising this point. We intended to indicate which clinical parameters in the Table were statistically significant. In response to this comment, we changed the footnote for p value from “a” to “*” in Tables 1–3.
- Line 151: Order the abbreviations according to the order of appearance of the term in the table.
Response: We thank the reviewer for their advice and changed the order of abbreviations accordingly (page 4, line 151).
- Table 2: The first column is not clear because of the alignment. The main titles: formal music training, number of musical training category and mean musical background level should be aligned to the left and the subtitles should be aligned all to the same level but a little more to the right. Please format all the tables in the same way.
Response: We agree with the reviewer’s advice and left aligned the first column in Table 2 as well as in other tables (Tables 1 and 2).
- Table 2: As stated on the M&M, all patients filled the questionnaire. Why was the number of patients different (n=42), only for the number of musical training?
Response: Of the total 49 patients (total study population), 42 experienced formal musical lessons. They were further analyzed by dividing them into a group that participated in one category and a group that participated in two or more categories to assess the difference in the level of their musical background.
- Figures 1 and 2: Authors present the analysis of the perception of music quality and the music elements factor by factor. In each case, we have a numeric variable (e.g. naturalness – unpleasant to pleasant), and two categorical variables 1. Elderly /non-elderly and 2. Preoperative/postoperative. Consequently, we need to answer these questions:
24-1. Were the preoperative values different in both groups? Sometimes, we observe that the values of the elderly group were better than non-elderly group
Response: We thank the reviewer for raising this point. The preoperative scores of non-complexity, melody, timbre, and lyrics in the elderly group was higher than those in the non-elderly group. However, in the present study, we focused on the postoperative changes in these variables in each group after CI surgery. Therefore, we compared postoperative values with preoperative ones using the statistical method for paired sample within each group (i.e. Wilcoxon signed rank test) as described in the Materials and Methods section.
24-2. Were the postoperative values different in both groups?
Response: Neither postoperative values in music quality or in music elements was significantly different between the elderly (P=0.83, P=0.24, P=0.11, P=0.18, P=0.07, P=0.20) and non-elderly groups (P=0.57, P=0.61, P=0.61, P=0.86, P=0.24, P=0.83).
- The goal of the study was to compare elderly and non-elderly CI users. So, we need to answer this question according to the objective. For example, the naturalness, the x axis could be preoperative and postoperative instead of non-elderly and elderly. In the case of the color of the boxplots, it could be non-elderly and elderly instead of preimplant and postimplant.
Response: We thank the reviewer for their advice. As mentioned in the response for number 24-1, in the present study, we focused on the postoperative changes of those variables in each group after CI surgery. Therefore, we compared postoperative values with preoperative ones using the Wilcoxon signed rank test. We believe that our results would be presented more effectively when the x-axis indicates the non-elderly and elderly groups, and the color of the boxplots indicate preoperative and postoperative status.
- Were the preoperative and the postoperative values different inside each group (non-elderly or elderly)? Was the deterioration of the musical appreciation more marked in the elderly than the non-elderly group?
Response: While there was no significant difference in the values of both music quality and music elements between pre-operation and post-operation in the non-elderly group, significant deteriorations in naturalness, clarity, similarity of sounds to music, non-complexity of music, and ease of following along were observed in the elderly group. The elderly group also showed significant postoperative decline in the perception of rhythm, melody, timbre, and lyrics. These were the main results of the present study.
- Sometimes I observe that the difference between the value pre and postoperative in the non-elderly group decrease less than the values observed in the elderly group (e.g. easy of following).
Response: We thank the reviewer for raising this point. However, there was no significant difference in values of both music quality and music elements, including ease of following (5.0 vs 5.0, P=0.59), between pre-operation and post-operation in the non-elderly group.
- Figure 1: The scale (y-axis) of all figures are 1, 3, 5, 7 with the exception of (f). Please use the same format for all figures
Response: We thank the reviewer for their advice. In response to this comment, we changed the scale of diagram (f) from “1,2,3,4,5,6,7” to “1,3,5,7” in Figure 1.
- Line 179 -180: Put the corresponding letter of the figure in the legend (e.g. postoperative scores for (a) naturalness, (b) clarity…
Response: We thank the reviewer for their advice. In response to this comment, we added the following letter legends in Figure 1 for improved consistency:
“(a), pleasantness; (b), naturalness; (c), clarity; (d), musicality; (e), non-complexity; (f), ease of following along” (page 6, lines 179–180).
- In figure 1, the colors are expressed as preimplant and postimplant. In the legend, authors write postoperative and preoperative, write the same term in both.
Response: We thank the reviewer for their advice. In response to this comment, we changed “preimplant” and “postimplant” to “preoperative” and “postoperative”, respectively in Figure 1 for consistency.
- Lines 198-199: This is not necessary to write (c), (d) or (e) twice.
Response: We agree with the reviewer’s advice and removed the redundant letter legends from the Figure legends.
- Both figures need to have the same format. In figure 1, we observe only (a), (b)… however, in figure 2 we observe (a) and a description.
Response: We thank the reviewer for their advice. In response to this comment, we matched the format by adding the description (headline) to each diagram in Figure 1, and added the following legends for consistency:
“(a), pleasantness; (b), naturalness; (c), clarity; (d), musicality; (e), non-complexity; (f), ease of following along” (page 6, lines 179–180).
- In the legend of the figure, authors have to state the meaning of the asterisk * p<0.05
Response: We thank the reviewer for raising this point. In response to this comment, we added “* p<0.05” to the Figure legends (lines 184 and 201).
Discussion section:
- Concerning the hearing performance, we observe in the table 1 that the postoperative WDS was similar in both groups. Was the music appreciation similar in both groups in the postoperative period? was it different? Was there a relationship between the speech recognition scores (objective measure) and the music appreciation (subjective measure)?
Response: We thank the reviewer for their comments. The score of postoperative music appreciation was not different between the two groups. In addition, we could not find any relationship between the speech recognition (objective measure) and the music appreciation (subjective measure) in each group.
- Line 249-250: Avoid the verb to be “instrumental and melodic composition are” use another verb instead of "are"
Response: We thank the reviewer for their advice. In response to this comment, we revised the statement in the Discussion section to:
“…which includes less complex instrumental and melodic compositions than those of other Korean pop genres” (page 8, lines 249–250).
- Line 261-262: Authors recommend to perform objective tests to assess musical appreciation, however there was used the reference 24 at the end of the sentence. I think that this reference could be removed.
Response: We thank the reviewer for their advice. The clinical assessment of music perception test has been widely used as an objective test for evaluating music perception in CI recipients Thus, we had cited it at the sentence. In response to this comment, we moved the citation to the proper position as follows:
“… Further research involving objective tests, such as the clinical assessment of music perception test [31], is needed to reinforce and validate our findings” (line 270-272).
Conclusion section:
36 and 37. Line 264: Avoid the use of the verb to be, use another verb such as suggest instead of are. At the end of the conclusion, add a little phrase concerning the perspectives or recommendations of this work to be studied in future trials.
Response: We thank the reviewer for their advice. Considering this comment together with the suggestion of the first reviewer, we revised our Conclusion statement to:
“Our results suggest that elderly individuals are susceptible to changes in music appreciation following CI surgery. Although expectations of CI outcomes may vary from person to person, for elderly patients who rely heavily on music listening to overcome negative psychosocial changes that occurs with aging, it is necessary to counsel them to set realistic expectations about music appreciation before surgery based on our results. In addition to the surgical counselling, training programs to improve rhythm perception should be included in music training for elder CI users during rehabilitation because a significant decrease in the perception of rhythm occurs contrary to non-elderly CI users” (pages 8–9, lines 274–281).
- Line 279: remove the work between parenthesis (research)
Response: We thank the reviewer for raising this point. In response to this comment, we removed the parenthesis from the sentence (page 9, line 290).
- The text in the funding paragraph is repeated in the acknowledgement. Use this text either in the funding or in the acknowledgements.
Response: We thank the reviewer for their advice. In response to this comment, we removed the funding information from the Acknowledgements section and changed the Acknowledgments statement to “None.” (page 9, line 299).
